

**Response of modern fluvial sediments to regional tectonic activity**
**along the Min River, Eastern Tibet**
Wei Shi[1,2], Hanchao Jiang[1,2,*], Hongyan Xu[1,2], Siyuan Ma[1], Jiawei Fan[1,2], Siqi
Zhang[1], Qiaoqiao Guo[1], Xiaotong Wei[1]
*[1]State Key Laboratory of Earthquake Dynamics, Institute of Geology, China*
*Earthquake Administration, Beijing 100029, China*
*[2]Lhasa Geophysical National Observation and Research Station, Institute of Geology,*
*China Earthquake Administration, Beijing 100029, China*
**Corresponding author**: Hanchao Jiang, E-mail: hcjiang@ies.ac.cn



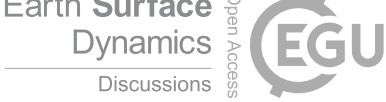

**Abstract**

The deposition of fluvial sediments in tectonically active areas is mainly controlled
by tectonics, climate, and associated Earth surface processes; consequently, fluvial
sediments can provide a valuable record of changes in regional climate and tectonic
activity. In this study, we conducted a detailed analysis of the grain–size distribution in
modern fluvial sediments from the upper Min River, Eastern Tibet. These data were
combined with regional information about climate, vegetation, hydrology,
geomorphology, lithology, and fault slip rate, and together indicate that modern regional
tectonic activity along upper Min River can be divided into three segments. Specifically,
fluvial sediments in the segment I are dominated by fine silts (<63 μm: 70.2%), agreeing
with a low–runoff and low–rainfall in this segment and revealing a windblown origin
influenced by the arid and windy climate. These observations are consistent with the
segment's low hillslope angle and low relief, all indicating weak activity along the
Minjiang Fault. The coarse–grained fraction (>250 μm) of fluvial sediments in the
segments II – III increases in a stepwise fashion (A = 6.2%, B = 19.4%, C = 33.8%)
downstream, although runoff and rainfall do not change significantly from segment II
to segment III. These patterns correlate well with an increase in both regional relief and
hillslope angles. Together, these observations imply that regional tectonic activity along
Maoxian–Wenchuan Fault becomes more pervasive downstream along the Min River.
Fluvial sediments in segment IV are well sorted and well rounded, which is expected
due to significant increases in rainfall and runoff in this segment. This study marks the
first development of a new and important research approach that can characterize



regional tectonic activity by analysis of fluvial sediments collected from tectonically
active regions, combined with regional conditions in geology and geography.
**Keywords:** Modern fluvial sediments; Grain-size analysis; Tectonic activity; Upper
Min River; Eastern Tibetan Plateau





## 1 Introduction


Tectonic geomorphology is a relatively young sub–discipline of geomorphology,
and has the major aim of unraveling interactions between tectonic activity, climate, and
Earth surface processes (Wobus et al., 2005; Owen, 2013). The grain size distribution
of river bed material, channel width, channel sinuosity, extent of alluvial cover,
lithology of bedrock, and hydraulic roughness are all potentially important variables
(Whipple, 2004; Whittaker et al., 2010). Thus, comprehensive amounts of data must be
collected in a wide range of field settings before the responses of these important
variables to climatic and tectonic forcings can be determined.
The topographic margin of the Tibetan Plateau (TP) along the Longmen Shan is
one of the most impressive continental escarpments in the world, and the land surface
rises westward over a horizontal distance of 40–60 km from the Sichuan Basin (500–
700 m elevation) to peak elevations exceeding 6000 m (Chen et al., 2000; Kirby et al.,
2000, 2008). Some studies have revealed common topographic features within river
channels in the eastern TP, namely, an upper low–gradient channel segment, a middle
steep–gradient channel segment, and a low–lying very steep channel segment, such as
in the Red River region in Yunnan Province (Schoenbohm et al., 2004) and the Min
River region in Sichuan Province (Kirby et al., 2003). However, it is important to note
that strong lithological contrasts along the length of a river can also cause the channel
steepness index to change at comparable magnitudes to those associated with large
gradients in rock uplift rate (Snyder et al. 2000; Stock and Dietrich 2003; Beek and
Bishop 2003; Whittaker et al., 2010). New data sourced from several localities record
an apparent narrowing of channel width in response to increased rock uplift rates in





rivers with large areas of bedrock (Whipple, 2004). This is consistent with the recent
proposition that river profiles straighten as aridity increase (Chen et al., 2019), as
observed along the upper Min River in the field. Generally, exposures of hard bedrock
often generate straight channels, which have low channel slopes and small sediment
loads (Schumm and Khan, 1971, 1972).

Vegetation density can modulate topographic responses to changing denudation

rates, such that the functional relationship between denudation rate and topographic
steepness becomes increasingly linear as vegetation density increases (Olen et al., 2016).
Recent studies indicate that the upper Min River has poor vegetation coverage and most
regions are fully exposed due to the strongly arid climate conditions (Jiang et al., 2015;
Xu et al., 2020; Shi et al., 2020; Wei et al., 2021; Zhou et al., 2021). Thus, hillslope
colluvium is the dominant sediment source to the upper Min River – especially in its
middle and lower segments (Zhang et al., 2021) – akin to those in drainage basins in
many arid regions worldwide (Clapp et al., 2002).

Tectonic activity influences the evolution of lacustrine sedimentary sequences by

affecting the provenance supply (Najman, 2006; Jiang et al., 2022). Frequent
earthquakes on the TP, as recorded by widely distributed soft sediment deformation
(Wang et al., 2011; Xu et al., 2015; Jiang et al., 2016; Zhong et al., 2019; Zhang et al.,
2021), caused repeated landslides that also represent another major source of sediment
into the upper Min River (Dai et al., 2011; Xu et al., 2012, 2013). These landslides
generated a large dust storm that deposited dust in nearby lakes (Jiang et al., 2014, 2017)
and exposed large quantities of fine–grained sediment that had accumulated on
mountain slopes, which were subsequently transported by wind to ancient lakes,
documenting these seismic events (Whittaker et al., 2010; Liang and Jiang, 2017; Shi
et al., 2022). This sedimentological process was recently recognized at Huojizhai, Diexi

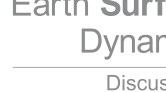

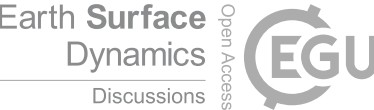

Town, following the historical earthquake at Diexi in 1933 (Wei et al., 2021).
Changes in hydrology and sediment flux are commonly regarded as climate
forcing (Wobus et al., 2010). The extent of alluvial cover is very limited throughout the
upper Min River Basin, which is demonstrated by similarity of zircon U–Pb ages in
lacustrine sediments and their nearby bedrock units (Zhong et al., 2017). As such, the
influence of occasional flood events should be considered over long time–scales
(Snyder and Whipple, 2003), as aridity precludes rainfall or fluvial undercutting as
being the trigger for such events.
The consistent climate coupled with systematic variations in lithology and rock
uplift rate along the Min Mountains allow comparison of channels that experience
different tectonic forcings (Duvall et al., 2004). Selective transport is the dominant
downstream fining mechanism in this region, although rates of selective transport in
sand–bed rivers are smaller than those in gravel–bed rivers (Frings, 2008).
Only a small volume of sediment collected from a river bed is needed to produce
a transformative understanding of the rates at which landscapes change (Blanckenburg,
2005). Study of these materials can reveal relationships between generation, transport
(Clapp et al., 2000, 2002), and mixing of sediment (Perg et al., 2003; Nichols et al.,
2005), under the help of the key topographic and/or lithologic features (e.g., relief, slope
angle, and substrate characteristics) (Riebe et al., 2000; Riebe et al., 2001; Matmon et
al., 2003a, b). In this study, we combine field observations, surveys, and analysis of
river sediments to determine hydraulic characteristics, and topographic and tectonic
information about bedrock channels in the upper Min River.

**2 Regional setting**
**2.1 Geographic and geologic settings**



Instrumental data collected after 1900 indicate that the TP has experienced strong
earthquakes clustered around the Bayan Kala Block from 1995 to the present day, which
are collectively known as the Kunlun–Wenchuan earthquake series (Deng et al., 2014).
The eastern TP is geomorphologically characterized by alpine valleys, and tectonic
activity is controlled by the Longmen Shan thrust belts, the Minjiang Fault, and the
Huya Fault (Fig. 1a). Frequent tectonic activities have led to numerous earthquakes and
landslides in this region (e.g., Zhang et al., 2003; Jiang et al., 2014; Li et al., 2015;
Liang and Jiang, 2017), such as the 1933 Diexi $M_s$ 7.5 earthquake, the 1976 Songpan
$M_s$ 7.2 earthquake, the 2008 Wenchuan $M_s$ 8.0 earthquake and the 2017 Jiuzhaigou $M_s$
7.0 earthquake. These earthquakes caused widespread damage at the surface in this
region. GPS–measured uplift rates in the Longmen Shan Fault zone reached 2–3 mm/a
over 10 years since 1999 (Liang et al., 2013). Thermochronological dating of zircon
and apatite indicated denudation rates of 1–2 mm/a in the Longmen Shan region during
the Late Cenozoic (Kirby et al., 2002).
The alpine valleys in the eastern TP reduce the preservation potential of
Quaternary sediments and expose large areas of bedrock. Bedrock outcrops within the
catchment region of the upper Min River are dominated by Silurian phyllite, quartz
schist, and Triassic phyllite, metamorphosed sandstone (Fig. 1a), which are easily
weathered and eroded into transportable debris (Zhong et al., 2019). Massive granites
are also exposed in the study area; in particular, the Neoproterozoic Pengguan complex
(U–Pb age of 859–699 Ma; Ma et al., 1996) (Fig. 1a) is mainly composed of
intermediate–acid intrusive rocks, with lesser amounts of basic–ultrabasic intrusive
rocks, volcanic rocks, volcanoclastic rocks, and greenschist facies metamorphic rocks.



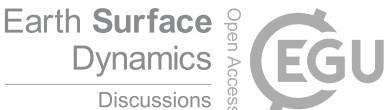

Sand (> 63 µm) in the study area was recently demonstrated to have been mainly
derived from local debris material, which itself is likely related to dust storms and loose
surface material produced by seismic activity (Jiang et al., 2017; Liang and Jiang, 2017).

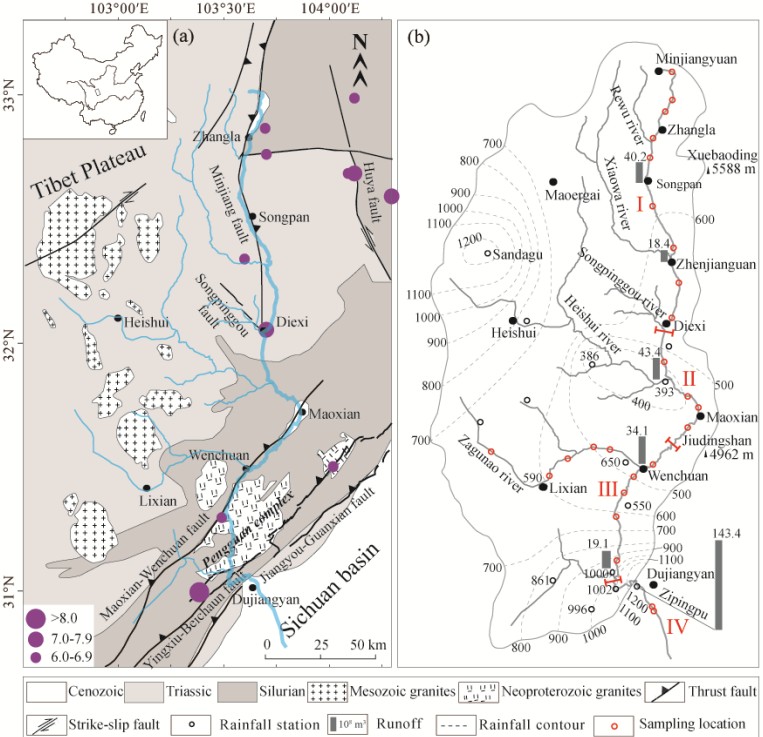

**Figure 1 (a)** Geological map and **(b)** precipitation distribution (Ding et al., 2014) for
the upper Min River basin. Seismic data are from the China Earthquake Data Center
(http://data.earthquake.cn/data).

The upstream channel of the Min River is ~340 km long (Li et al., 2005; Ding et
al., 2014), nearly oriented N–S (Fig. 1b), and erodes the hinterland of the TP via
formation of gullies and valleys. The Min River valley is typically steep, narrow and
deepening downstream with an incision depth of 300–1500 m (e.g., Li et al., 2005;
Zhang et al., 2005). The slopes on both sides of the study area are between 18° and 45°,





and the vertical aspect ratio of the valley is 5.5–12.6 ‰ (Zhang et al., 2005).
Constrained by the specific landforms of the alpine valleys, the wind direction in the
study area is generally SSW/NNE, roughly consistent with the strike of local valleys
(Liu, 2014). The Min River valley exhibits high wind speeds in April (average 4.9 m/s)
and low speeds in July (average 3.7 m/s). Wind speed is generally < 4 m/s before noon
and > 4 m/s after noon, and normally peaks approximately 8–10 m/s at around 16:00
(Liu, 2014). The highest instantaneous wind speed recorded in the study area was 21
m/s (Liu, 2014).
The upper reaches of the Min River are located in a transition zone on the TP
where wet monsoonal climate changes to a high–elevation cold region. In this region,
mean annual precipitation (MAP) ranges from 400 mm to 850 mm, and precipitation is
dominant (>75%) during the rainy season (May–October) (Ding et al., 2014). It is
noticeable that orographic rain along the eastern TP generates two storm areas centered
around Sandagu and Zipingpu (Fig. 1b). Statistical analyses of precipitation data from
1982 to 2007 show that the MAP within these regions is higher than 1200 mm (Ding et
al., 2014).
Regional vegetation has clear vertical zonation, which mainly consists of small–
leaf, arid shrubs at 1300–2200 m a.s.l., mixed broadleaf–conifer forests, evergreen and
deciduous broad–leaved mixed forests at 2000–2800 m a.s.l., *Picea* and *Abies* forests
at 2800–3600 m a.s.l., and alpine shrubs and meadows at > 3600 m a.s.l. (Ma et al.,
2004; Zhang et al., 2008; Wei et al., 2021; Xu et al., 2020). There are two key factors
that influence vegetation distribution and ecological conditions in the study area: the
arid and windy climate, which has a large temperature difference between day and night,
and tectonics activity characterized by frequent earthquakes (Lin, 2008; Wang et al.,
2011). For example, strong earthquakes often induce landslides that can destroy



vegetation cover in the study area (Xu et al., 2012, 2014). Both of these factors lead to
fragility in landscape and vegetation cover.
**2.2 Segmented characteristics of the Min River**
The topographical and geomorphological characteristics, and fault and vegetation
distribution patterns of the upper Min River allow it to be subdivided into four segments:
I, II, III, and IV (Fig. 1b).

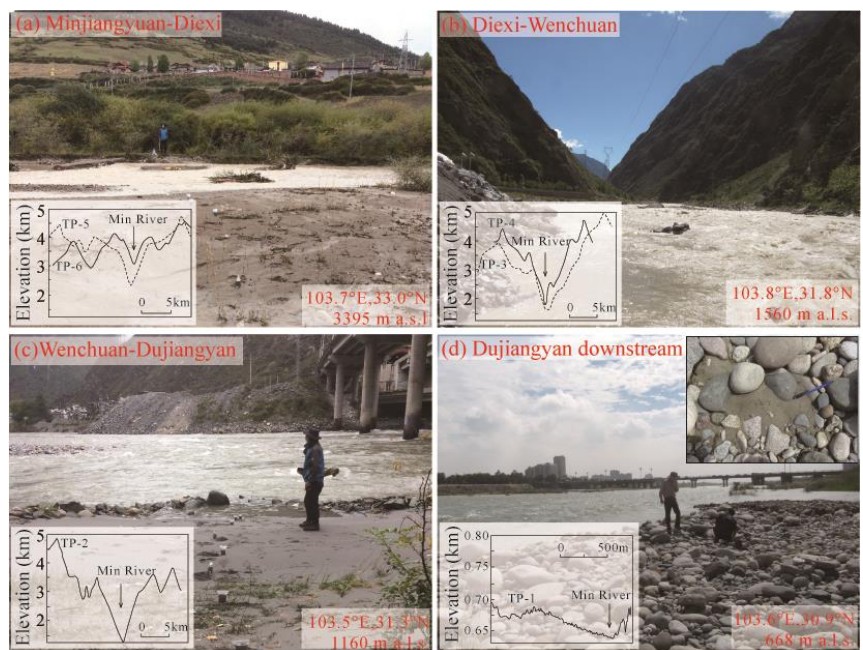

**Figure 2** Photograph of field sampling sites in the upper Min River. The locations of
cross–sections though the Min River valleys (Zhang et al., 2005) are shown in Fig. 7c.

Segment I is the Minjiangyuan – Diexi segment (3460–2190 m a.s.l.). The riverbed
in this segment is directly connected with one side of the Min Mountain and has a valley
bottom width of 200–1000 m (Zhang et al., 2005) (Fig. 2a). Downstream from the
Minjiangyuan, valley bottom width narrows markedly and is only 200–300 m in
Zhenjiangguan – Diexi segment (Zhang et al., 2005). The relative relief of the Min



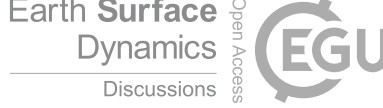

Mountain increases significantly from Minjiangyuan to Diexi along the Min River,
especially from the Zhenjiangguan to Diexi (Zhang et al., 2005). The vegetation
coverage along this segment gradually deteriorates, with *Picea*, *Abies*, shrubs, and herbs
in the Minjiangyuan – Songpan segment, but only a small number of shrubs and herbs
in the Songpan – Diexi segment. Bedrock is widely exposed in the lower part of the
segment. In this region, the monthly maximum wind speed reaches 15.4 m/s in Songpan.

Segment II is the Diexi–Wenchuan segment (2190–1470 m a.s.l.). The valley

bottom width in this segment continuously decreases to 200–300 m (Zhang et al., 2005),
and the Min Mountains always occur in direct contact with the riverbed of the Min
River (Fig. 2b). The longitudinal slope (12.6‰) reaches its maximum regional value
near Diexi (Zhang et al., 2005). The regional vegetation coverage is mostly sparse and
the bedrock is naked.

Segment III is the Wenchuan–Dujiangyan segment (1470–900m a.s.l.). The valley

bottom width in this segment widens to about 200–500 m (Zhang et al., 2005) (Fig. 2c)
and regional vegetation cover increases compared to segment II. In particular, the
hillside around the Zipingpu Reservoir is covered with thick broad–leaved trees and
herbs. The monthly maximum wind speed in Lixian is 14.0 m/s.

Segment IV is the Dujiangyan– segment (900 – 630 m a.s.l.). This segment flows

into the interior of the Sichuan Basin, where it has flat geomorphological features (i.e.,
the riverbed width is greater than 300 m; Fig. 2d), and then transitions into the middle
reach of the Min River. The monthly maximum wind speed in Dujiangyan is 13.8 m/s.

**3 Materials and methods**
**3.1 Field sampling and grain–size analysis**



A ~265 km transect along the upper Min River was conducted during October
2017, starting in the eastern TP (Minjiangyuan, 33°01′59″N, 103°42′42″E; 3462 m a.s.l.)
and ending in the Sichuan Basin (Dujiangyan, 30°56′25″N, 103°38′14″E; 634 m a.s.l.)
(Fig. 1b). A total of 181 river samples were collected for grain–size analysis at 25 sites
(Table S1). Sampling sites were selected from exposed, freshly–developed depositional
sequences that occurred close to the active channel and its margins (Fig. 2). Voluminous
bedrock gravel occurs around the sampling sites (Fig. 2). To ensure sample consistency
associated with uniform flow regimes, each sample was collected at a depth of 0–0.2 m
from different places within each sampling sequence. All locations were carefully
chosen to avoid contamination from riverbank materials or from anthropogenic
reworking.
Grain–size analysis was conducted using a Malvern Master–sizer 3000 laser
grain–size analyzer at the State Key Laboratory of Earthquake Dynamics, Institute of
Geology, China Earthquake Administration in Beijing, China. About 0.5 g of sediment
was pretreated with 20 ml of 30% $H_2O_2$ to remove organic matter and then with 10 ml
of 10% HCl to remove carbonates. About 300 ml of deionized water was added, and
the sample solution was kept for 24 h to rinse acidic ions. The sample residue was
dispersed with 10 ml of 0.05 M $(NaPO_3)_6$ on an ultrasonic vibrator for 10 min before
grain–size measurements. For each sample, the grain–size analyzer automatically
outputs the median diameter (Md) and the percentages of each size fraction, with a
relative error of less than 1%. Magnetic susceptibility (SUS) was measured using a
Bartington MS2 susceptibility meter.





**3.2 Y values**

Mean grain size (Ms), standard deviation (σ), skewness (Sk), and kurtosis ($K_G$) are commonly used to discriminate between different depositional processes and environments. Sahu (1964) distinguished aeolian processes from those that operate in a littoral environment by using the following equation:

$$Y = -3.5688\,Ms + 3.7016\,\sigma^2 - 2.0766\,Sk + 3.1135\,K_G \qquad (1)$$

Here, Y values less than −2.74 indicate an aeolian provenance and Y values greater than −2.74 indicate a hydrogenic provenance (Sahu, 1964). Calculated Y values for lacustrine sediments (Jiang et al., 2017, 2014), red clay, and loess–paleosol deposits (Wu et al., 2017; Lu and An, 1999) are less than −2.74, indicating an aeolian provenance.

**3.3 End–member analysis**

Numerical unmixing of grain–size distribution data into constituent components, known as end–member analysis (EMA), can yield valuable information about transport dynamics (Weltje, 1997; Paterson and Heslop, 2015; Jiang et al., 2017). According to the principle that the end–member number (EM) should be as small as possible (Weltje et al., 1997), several EMs obtained by end–element analysis imply that numerous dynamic mechanisms occurred during formation of these deposits. Generally, larger values of EMs correspond to a stronger transport capacity, which itself indicates different provenances (Vandenberghe, 2013; Dietze et al., 2014; Jiang et al., 2017). For instance, the peak values of EMs in Lixian lacustrine sediments were concentrated at 10 μm ($EM_1$) and 40 μm ($EM_2$), and so reflect the background deposition of dust and locally sourced deposition transported by ambient wind, respectively (Jiang et al., 2017). We analyzed the Min River samples using the AnalySize software for processing and



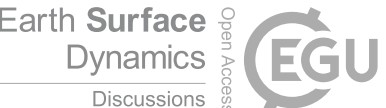

unmixing grain–size data (Paterson et al., 2015), with parameters selected from the
generalized Gaussian skewness model (SGG) (Egli, 2003).
**3.4 Analysis of C–M and F–M diagrams**
The analysis of C–M and F–M diagrams is useful to interpret sediment transport
dynamics (Passega, 1957; Singh et al., 2007). In these diagrams, C is the coarsest
percentile of the grain–size distribution in samples (one percentile), and M is the median
diameter of the grain–size distribution, which are both indicators of the maximum and
average transport capacity, respectively (Passega, 1957; Singh et al., 2007; Bravard et
al., 2014). In addition, F represents the percentage of fractions finer than 125 μm (Singh
et al., 2007). All values are plotted on a logarithmic scale, which produces specific
patterns for distinct reaches (Singh et al., 2007; Bravard et al., 2014). A C–M diagram
(Fig. S1) has the following sections: NO, rolling; OP, rolling with some grains
transported in suspension; PQ, graded suspension with some grains transported by
rolling; QR, graded suspension; RS, uniform suspension; and T, pelagic suspension
(Passega, 1957; Bravard and Peiry, 1999; Bravard et al., 2014).

**4 Results**
**4.1 Characteristics of grain–size and SUS**
The median grain size (Md), five grain-size fractions (0-2 μm, 2-20 μm, 20-63 μm,
63-250 μm, >250 μm), SUS and Y values of the Min River sediment can be divided
into four categories (Fig. 3), which correspond to the different segments (I – IV) defined
above. The average values of Md increased significantly at Diexi (from 31.0 μm to 80.8





μm) and Wenchuan (from 49.3 μm to 170.2 μm), and decreased slightly at Dujiangyan

(from 220.4 μm to 119.2 μm). The variations at these three sites are the most significant

within the whole river (Table 1, Fig. 3).

**Table 1** Statistics for grain–size fractions in the upper Min River.

| Segments | Md (μm) | Percentage composition / (%) | | | | | SUS |
|---|---|---|---|---|---|---|---|
| | | 0–2 μm | 2–20 μm | 20–63 μm | 63–250 μm | >250 μm | |
| I | 31.0 | 2.8 | 40.3 | 27.1 | 23.7 | 6.2 | 11.6 |
| II | 80.8 | 0.4 | 25.3 | 20.3 | 34.6 | 19.4 | 11.3 |
| III | 170.2 | 0.3 | 20.0 | 13.9 | 31.9 | 33.8 | 193.5 |
| IV | 145.2 | 0.5 | 13.0 | 9.5 | 59.5 | 17.5 | 251.8 |

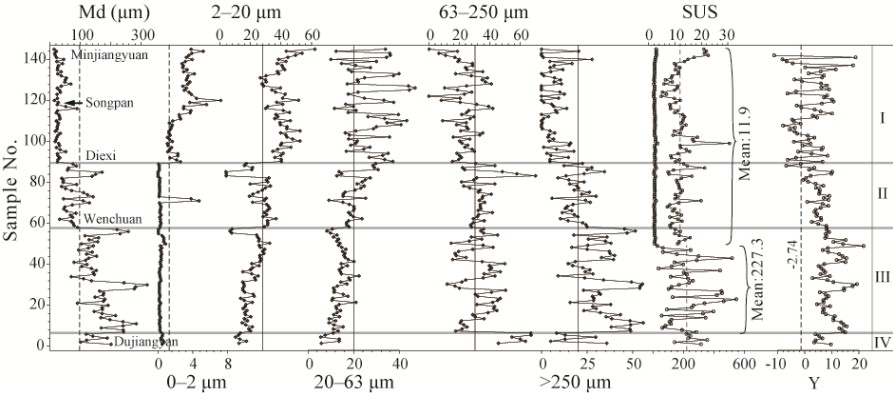

**Figure 3** Variation of grain–size components and river sediment parameters from the

upper Min River.

Along the upper Min River downwards, the mean proportion of the 2–20 μm (I =

40.3%, II = 25.3%, III = 20.0%, and IV = 13.0%) and 20–63 μm fractions (I = 27.1%,

II = 20.3%, III = 13.9%, and IV = 9.5%) exhibit a stepwise decrease (Table 1, Fig. 3).

The 63–250 μm fraction exhibits a sharp increase from segment I (23.7%) to II (34.6%)



and from segment III (31.9%) to IV (59.5%), but a relatively minor change from
segment II (34.6%) to III (31.9%) (Table 1, Fig. 3). The > 250 μm fractions exhibit a
stepwise increase between segments I, II, and III (6.2%, 19.4%, and 33.8%,
respectively), and a significant decrease from segment III (33.8%) to IV (17.5%) (Table
1, Fig. 3). Measured SUS values remained low in segments I (5.3–30.6, with a mean of
11.6) and II (7.1 to 21.2, with a mean of 11.3), but were significantly higher in segment
III (9.9–546.5, with a mean of 193.5) and reached consistently high values in segment
IV (142.1–356.5, mean: 251.8) (Table 1, Fig. 3).
**4.2 End–member analysis**

Three end–members (EMs) ($R^2 = 0.93$) were identified in the Min River samples

(Fig. 4) with peaks of 21.2 μm (58.0%), 185.8 μm (24.2%), and 351.7 μm (17.8%).
Along the upper Min River downwards, these three EMs show clear stepwise changes
between segments (Fig. 5). $EM_1$ shows a stepwise decrease (I = 82.5%, II = 53.1%, III
= 38.6%, and IV = 23.7%), corresponding to the sum of the 2–20 μm and 20–63 μm
fractions (Figs. 3, 5). $EM_2$ shows a sharp increase from segment I (13.1%) to II (31.4%)
and from segment III (27.1%) to IV (67.4%), and a relatively smaller change from
segment II (31.4%) to III (27.1%), corresponding to the 63–250 μm fraction. By contrast,
$EM_3$ corresponds to the >250 μm fraction (Figs. 3, 5) and shows a stepwise increase
between segments I, II, and III (4.4%, 15.5% and 38.6%, respectively), and a significant
decrease from segment III (38.6) to IV (23.7%).



Earth **Surface**
**Dynamics**
Discussions

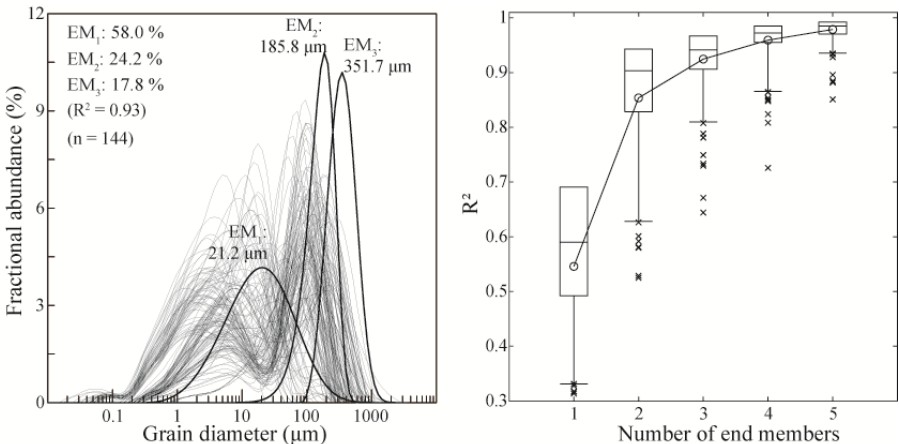

**Figure 4** End–member analysis model of fluvial sediments from the upper Min River.

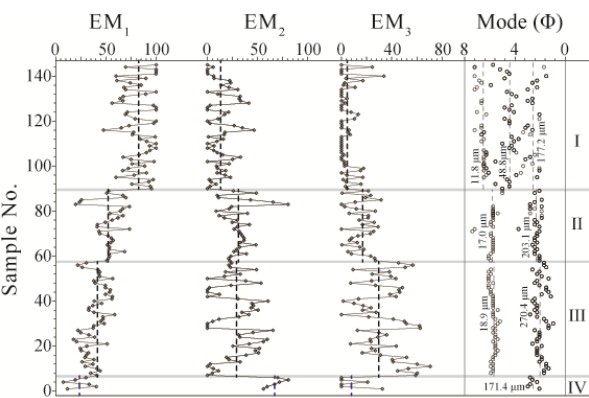

**Figure 5** Variability of three EMs and their mode values from the upper Min River. The
fractional abundance (>1%) of the peak in the grain–size frequency distributions were
extracted after consideration of a 1% instrumental error. Black and gray circles
represent the main and secondary peak modal values, respectively.

**4.3 Characteristics of the grain–size frequency distribution**

The grain–size frequency of river samples from segment I has a discrete

distribution (Fig. S2) with three mode values at ~11.8 μm, ~48.8 μm, and ~177.2 μm.
The main mode value of segment I occurred in the ~48.8 μm portion. The grain–size





frequency distribution for segments II and III is strongly bimodal (Fig. S2), with the
major and minor mode values at ~203.1 μm and ~17.0 μm for segment II, and ~270.4
μm and ~18.9 μm for segment III. The grain–size frequency distribution for segment
IV is unimodal (Fig. S2) with a mode value of ~171.4 μm.

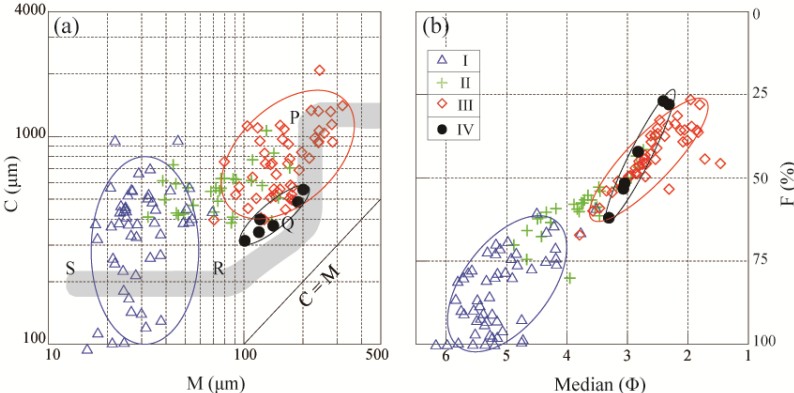

**Figure 6** C–M and F–M distributions of samples collected from the four studied
segments of the upper Min River.

**4.4 C–M and F–M diagrams**

On a C–M diagram for the Min River, samples from segment I are completely

separate from those collected from segments III and IV. Most samples in segment II
overlap with those of segment III (Fig. 6a). Among them, the M value of segment I
(13.9–89.8 μm) mainly belongs to the RS section (Fig. 6a), although the C values
exhibit a large variation between 54.8 μm and 964.3 μm. Samples from segment II are
distributed throughout the P–Q–R sections (Fig. 6a), have C values of 383.5–1066.0
μm, and M values of 32.2–171.4 μm. Samples from segment III are concentrated in the
PQ section (Fig. 6a), have C values of 396.9–2083.8 μm, and M values of 70.3–319.1

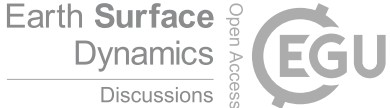

μm. Samples in segment IV plot close to the RQ section and are distributed parallel to
the C = M line (Fig. 6a). Samples collected from segments of the Min River show
similar distribution features in F–M diagrams to those shown in C–M diagrams (Fig.

6).


**5 Discussion**
**5.1 Dynamic and provenance implications of fluvial sediments**
Grain–size fractions, EMs, and mode values in different segments along the upper reach
of Min River reflect the distinct provenance and transport dynamics of fluvial sediments
(McKinney and Sanders, 1978; Sun et al., 2002, 2004; Sun et al., 2007; Dietze et al.,
2014; Vandenberghe, 2013). The $EM_1$ in segment I reaches a proportion of 82.5%,
which corresponds to the fine particle components (<63 μm fractions). Previous studies
have indicated that fractions with sizes of 10–40 μm represent background particles and
regional dust that have been transported by wind (Jiang et al., 2014, 2017), which
contribute 51±11% and 42±14% of the lacustrine sediments across the TP, respectively
(Dietze et al., 2014). Therefore, the $EM_1$ (fine–grained fractions) in segment I probably
have an aeolian provenance. This inference is supported by five separate lines of
evidence: 1) Md varies within the narrow range 13.9–89.8 μm (Fig. 3), although the C
values fluctuate widely between 54.8 μm and 964.3 μm (Fig. 7); 2) the distribution of
samples in an RS section in a C–M diagram (Fig. 6) reflects uniform suspension, which
likely requires transportation by ubiquitous and strong wind (Fig. S1, Passega, 1957);
3) nearly half of the samples (i.e., 22 out of 55) have Y values of less than –2.74, which



is indicative of an aeolian origin (Sahu, 1964); 4) loess deposits are widely distributed
in the study area, especially from Diexi upstream (Fig. S3) (Liu et al., 2013; Shen et al.,
2017) and may represent a voluminous source of dust particles; and 5) the study area
has a high mean altitude of 2840 m, and the monthly maximum wind speed can reach
15.4 m/s, which would allow for strong aeolian transport.

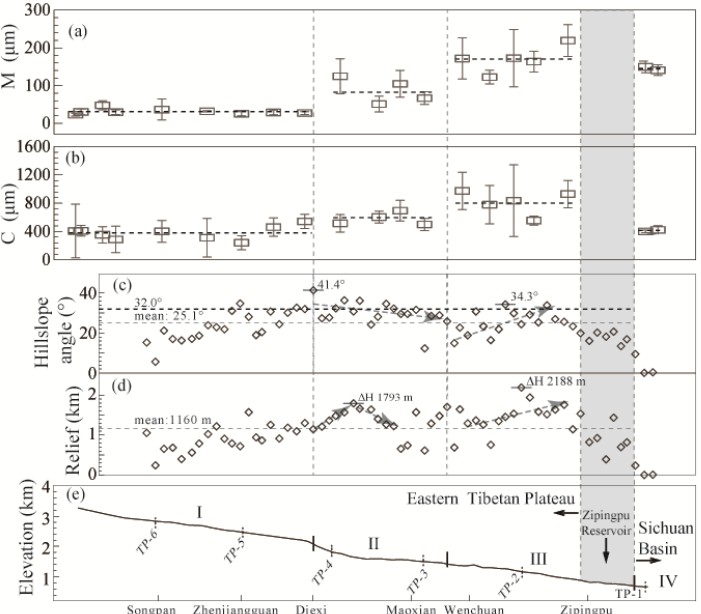

**Figure7** Variation characteristics of **(a)** M and **(b)** C values of the grain–size index. **(c)**
Riverbed base–level and the position of the cross–section of the upper Min River
(Zhang et al., 2005). **(d)** Hillslope angle and **(e)** local relief along the upper Min River.
A 4*4 km grid was delineated along the upper Min River (~260 km). The highest
ridgeline and riverbed height in the grid were extracted from a DEM map, and the local
relief was then obtained by calculating the highest ridgeline minus the riverbed height.
The hillslope angle was obtained by solving for tan (local relief/slope length).





The EM$_2$ in segment IV reaches the highest value (185.8 μm: 67.4%) recorded in
the whole sequence and corresponds to the 63–250 μm fraction (59.5%), which is
consistent with previous studies having shown that fluvial deposits are composed
mainly of a medium–sand component (modal size: 200–400 μm) (Middleton, 1976;
Tsoar and Pye, 1987; Bennett and Best, 1995; Dietze et al., 2014). In the C–M diagram,
sample data that lie close to the C = M line reflect the suspension transport of riverbed
sediments (Fig. 6a) (Singh et al., 2007; Passega, 1957). In addition, the single peak
mode (Fig. S2d) of segment IV represents a single river transport process and
sedimentary environment (McKinney and Sanders, 1978), and the small size range of
the grain–size frequency distribution also reflects a well–sorted product that was
deposited by fluvial action (Sun et al., 2002). Therefore, the EM$_2$ mainly reflect typical
fluvial sediments.
EM$_3$ corresponds to the coarsest grain–size components (>250 μm) and has the
highest value (351.7 μm: 38.6%) of the whole sequence in segment III. The maximum
values of C and M (Figs. 7a, b) in segment III indicate that it had the highest transport
capacity (Passega, 1957; Singh et al., 2007; Bravard et al., 2014). Therefore, EM$_3$
represents the local sedimentary component that was locally transported over short
distances (Dietze et al., 2014; Jiang et al., 2014, 2017). The distribution characteristics
of samples from segment III in the PQ section (Fig. 6a) indicate that rolling and jumping
transportation processes dominated (Passega, 1957). Meanwhile, the SUS values in
segment III increase to abnormally high values (28.5–546.5, with a mean of 227.3)
abruptly near to exposures of the Pengguan complex (Fig. 1a), although lower SUS





values occur in the surrounding area (Zagunao River: 9.1–114.1, with a mean of 34.1,
Fig. S4; Zipingpu reservoir: 5–60, Zhang et al., 2019; and segments I and II: 5.3–30.6,
mean 11.5, Fig. 3). The precipitation in segment III is generally low (400–700 mm/a)
and only significantly increases near to the Zipingpu reservoir (1200 mm/a), so that the
sedimentary changes were muted until 2 years after the Wenchuan earthquake (Zhang
et al., 2019) (Fig. 1b). In addition, the mean grain size in segment III (170.2 μm)
increases before the Zagunao River (mean of 83.1 μm, Fig. S4) joins the Min River (Fig.
1b. 3) and contribution from the Zagunao River can be precluded. Therefore, the
abnormally high grain size and SUS values in segment III are likely caused by a local
provenance change.
**5.2 Climate controlled fine–grained fluvial sediments**
The windy and semi–arid climate in the study area is responsible for more fine
particle components (EM$_1$) in segment I (Jiang et al., 2014), which caused EM$_1$ to
gradually decrease downstream as the wind weakens (Fig. 5). The relatively low
precipitation (400–700 mm/a) and low runoff (18.4–43.4 × 10$^8$ m$^3$) (Fig. 1b) in segment
I reflect the limited transport capacity of the river, and the angular gravels on the
riverbed also indicate weak scouring, which preserves more fine–grained components
(EM$_1$) in fluvial sediments. Segment I developed along the Minjiang Fault (Fig. 1a),
which has a low slip rate (0.30–0.53 mm/a, Kirby et al., 2000; Zhou et al., 2000, 2006;
Tan et al., 2019) and therefore a weak influence on local provenance supply (Jiang et
al., 2014, 2017). In addition, the wide riverbed (Fig. 2a), relatively low hillslope angle,
and local relief in the Minjiangyuan–Songpan segment (Figs. 7d, e) causes *in situ*



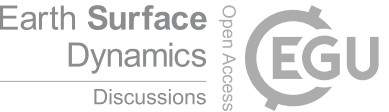

retention of locally sourced coarse components. Therefore, EM$_2$ and EM$_3$ make only a
minor contribution to the fluvial sediments in segment I.

Segment IV is located inside the Sichuan Basin and is completely unaffected by

alpine valleys in the eastern TP. It is characterized by a wide and flat geomorphological
surface (Fig. 2d). The significant downstream increase in precipitation and runoff in the
Zipingpu reservoir (Fig. 1b) indicates that fluvial action was the main control on
sediment transportation in segment IV.

### 5.3 Coarse–grained deposits controlled by tectonism

Fluvial sediments coarsen at the transition between segments I and II, highlighting

an increase in EM$_2$ and EM$_3$ content, and a higher M value (Figs. 3, 7). This locality
occurs at intersection of the Minjiang Fault and the Songpinggou Fault (Fig. 1a), which
was the epicenter of the Diexi *Ms* 7.5 earthquake in 1933 (Chen et al., 1994; Ren et al.,
2018). As a result, the outcropping bedrock was severely damaged and so provided new,
fresh, and local sediment sources (EM$_3$). Downstream from Diexi, field surveys exhibit
that the altitude decreases by 400 m over a horizontal distance of 20 km, such that the
longitudinal slope of the riverbed (12.6‰, Fig. 7c, Zhang et al., 2005) and the hillslope
angle (41.4°, Fig. 7d) are highest in this region when compared to the entire study area,
which imply a higher of rivers incision rates forced by active tectonics (Zhang et al.,
2005; Whittaker et al., 2007a). These remarkable changes of geomorphology
correspond well to a twofold increase in erosion coefficients that occur within 15 km
of major faults in the eastern TP (Kirkpatrick et al., 2020) and more intense denudation
at the location of seismogenic faulting along high–relief plateau margins (Li et al.,
2017). The narrower valley and direct contact between the riverbed and hillside on
either side in segment II (Fig. 2b) provide favorable conditions for rolling and jumping





transportation of sediment along the hillslope. In addition, the rapid rising of the base–
level of the Min River in segment II enhances the river's cutting and transport capacity
(Merritts and Vincent, 1989; Stokes et al., 2002; Cheng et al., 2004; Whittaker et al.,
2007a; Boulton et al., 2014).

Measured $EM_3$ rapidly reaches its maximum fluctuation range in segment III (Fig.

5), likely due to the maximum transport force (C value) in the area (Fig. 7). The regional
precipitation in segment III is low (400–700 mm/a) and only significantly increases
near to the Zipingpu reservoir (1200 mm/a) (Fig. 1b). From a tectonic perspective, the
Maoxian–Wenchuan Fault, with a large dextral slip rate (1.0–3.8mm /a; Chen and Li,
2013; Wang et al., 2017) and a large vertical slip rate (~1–2 mm/a; Liu et al., 2015),
mainly controls the distribution of segment III (Fig.1). Previous studies have shown that
the Maoxian–Wenchuan Fault occurs a band of maximum exhumation along the eastern
Longmen Shan Fault zone since the late Miocene (Tan et al., 2019). Therefore, rapid
regional uplift and denudation (Kirby et al., 2002; Liang et al., 2013) not only generated
a larger hillslope angle (mean value of 24.9°) and the highest local relief (2188 m), but
also provided widespread source of fresh, coarse–grained, and local sediment
(Whittaker et al., 2007b, 2010) in segment III. The significant coarsening of fluvial
sediment at the beginning of segment III indicates the catchments undergoing a transient
response to tectonics are associated with significant volumetric export of material
(Whittaker et al., 2010). Moreover, the PQ distribution of segment III samples in the
calculated C–M diagram (Fig. 4) shows the importance of rolling and jumping transport
mechanisms (Passega, 1957), which correlate with the steep landform features in
segment III (Fig. 2c). Exposures of hard Mesozoic granites instantaneously provide a





local source of coarse components, and thus correspond to the maximum M and C
values. Although regional climate generally has a weak influence on the supply of
coarse particles, the concentrated distribution of particles within the calculated grain–
size frequency distribution (Fig. S2c) indicates that fluvial action played an effective
role in sorting local sediment sources (Sahu, 1964; Sun et al.,2002; Frings, 2008). The
persistent occurrence of the coarsest grain–size cross the segment III responds to the
fact that the catchments crossing faults maintain their high slip rate over time, which
exhibits a sharp contrast to that of segment I.
Generally, a large earthquake is followed by a period of enhanced mass wasting
and fluvial sediment evacuation (Hovius et al., 2011; Wang et al., 2015). The Wenchuan
*M*s 8.0 earthquake in 2008 caused severe geomorphological damage in region, and the
annual average suspended sediment flow in regional rivers increased by a factor of 3–
7 following the earthquake. The river recovered to its pre–earthquake level just 1.2 ±
0.9 years later (Wang et al., 2015), however, over 70% of the co–seismic debris has
stabilized in place along the hillslopes during the following decade (Dai et al., 2021)
and will take 370 years to remove (Wang et al., 2017). As such, we believe that co–
seismic debris generated by the Wenchuan earthquake in 2008 had negligible influence
on our sample collection campaign conducted in 2017.
**5.4 Geomorphic morphology reveals tectonic activity**
Alpine valleys characterize the landscape of the upper reaches of the Min River in
the eastern TP (Figs. 2, 7) and have an incision depth of 300–1500 m (Li et al., 2005;
Zhang et al., 2005) (Fig. 6a). In segment I, hillslope angles and local relief gradually





increase downstream along the Minjiang Fault from 5° to 34.8° and 243 m to 1572 m,
respectively (Figs. 7d, e). However, these changes seem a little contradict with the
consistent high proportion of fine–grained background dust in the fluvial sediments of
segment I (Figs. 3, 5), which is an open and interesting question. The consistent
precipitation and runoff rates explain the calculated consistency in transport power, as
defined by unchanging values of C and M (Fig. 7). We note that the longitudinal slope
of the riverbed (6.7–7.6‰, Fig. 7c; Zhang et al., 2005) in segment I steadily changes as
altitude decrease from 3460 m to 2190 m; therefore, gradual steepening of the landscape
is likely a response to enhance river–related erosion (Merritts and Vincent, 1989; Stokes
et al., 2002; Cheng et al., 2004). The high vegetation density in the Minjiangyuan–
Songpan region is also probably modulated by the lower topographic slope (Figs. 2a, 7)
(Olen et al., 2016). These are consistent with generally weak activity of the Minjiang
Fault (Kirby et al., 2000; Zhou et al., 2000, 2006; Tan et al., 2019).

In segment II, the hillslope angle (12.3–41.4°, with a mean of 30.1°) is generally

steeper than the average for the whole study area (25.1°), and the highest angles (41.4°)
far exceed the stability threshold of ~32° for landslide denudation, which suggests that
landslide–dominated hillslope denudation has kept pace with the rates of rock uplift and
valley incision in segment II (Burbank, et al., 1996; Montgomery and Brandon, 2002;
Clarke and Burbank, 2010; Wang et al., 2014). Along the studied transect, local relief
in segment II initially increases and then decreases (Fig. 7c), and the flow direction of
the Min River also changes from roughly N–S to NW–SE (Fig. 1a). The lithology in
segment II changes from Triassic to Silurian (Fig. 1a), and seismic activity transitions



from the Minjiang Fault to the Maoxian–Wenchuan Fault. Given that segment II records
the lowest annual rainfall in the study area (<500 mm/a, Fig. 1), this transformation of
tectonic activity and lithology likely plays a dominant role on fluvial erodibility (Selby,
1980; Stokes et al., 2008; Whittaker et al., 2007a; Zondervan et al., 2020), and
influences changes in regional geomorphology and river drainage.

Hillslope angles (14.9°–34.3°, with a mean of 24.9°) and local relief (689–2188 m,

with a mean of 1463 m) in segment III exhibit a general increase along the Maoxian–
Wenchuan Fault (Figs. 1, 7), although they differ from the increasing trends shown in
segment I. For example, the highest local relief encountered throughout the entire
sequence occurs in segment III, although its mean hillslope angle (24.9°) is lower than
the mean value (25.1°) for the entire sequence (Fig. 7). In addition, precipitation and
runoff only show a significant increase adjacent to the Zipingpu reservoir (Fig. 1). We
note that the regional bedrock in segment III is dominated by hard Mesozoic granites
of the Pengguan complex (Fig. 1a), and that the Maoxian–Wenchuan Fault is situated
on the zone of maximum exhumation along the Longmen Shan fault zone (Tan et al.,
2019). Therefore, the higher local relief along segment III indicates that active
Maoxian–Wenchuan Fault (Tan et al., 2019) caused enhanced rock uplift and valley
incision (Whittaker et al., 2007a; Tan et al., 2019), which accounts for the largest
transport forces (C values, Fig. 7) and the coarsest local components (EM$_3$, Fig. 5) in
this section. Nevertheless, a decrease in the mean hillslope angle within segment III
may be attributed to hardening of the exposed bedrock of the Pengguan complex rather
than weakening of tectonic activity along the Maoxian–Wenchuan Fault. Even if
shortening rates are generally slow in the eastern TP (Densmore et al., 2008; Zhang,
2013) and satellite data may be equivocal, grain-size analysis of fluvial sediments
combined with topographic analyses can help guide the identification of regional
tectonic activity effectively (Schoenbohm et al., 2004; Kirby et al., 2003; Tan et al.,

2019).


**6 Conclusion**
Grain–size analysis was conducted on modern fluvial sediments of the upper Min
River and this information was integrated with vegetation, hydrology, geomorphology
(local relief and hillslope) and geology (fault and lithology) data to extract regional
climate and tectonic signals in the eastern TP. This procedure identified three segments
of tectonic activity along the upper Min River. The Minjiang Fault, situated in the
Minjiangyuan–Diexi segment, generally shows weak seismic activity. Two segments of
the fault from Diexi to Wenchuan and from Wenchuan to Dujiangyan show enhanced
phase of regional tectonic activity, although the segment from Dujiangyan to the
Sichuan basin records almost no evidence of tectonic activity.

**Data availability**
Data are available in the figshare database
(https://doi.pangaea.de/10.6084/m9.figshare.17111402).

**Author contributions**



The paper was written by WS and HCJ with major contributions by HYX. SYM
got geomorphic data. WS, HYX and SQZ participated in field surveys and sample
collection. SQZ, JWF and XTW conducted laboratory tests and interpreted the results.
All authors reviewed and approved the paper.

**Competing interests**
The contact author has declared that neither they nor their co-authors have any
competing interests.

**Acknowledgements**
This study was supported by the National Nonprofit Fundamental Research Grant
of China, Institute of Geology, China Earthquake Administration (IGCEA2126 and
IGCEA1906).

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
