# Peer review of "Response of modern fluvial sediments to regional tectonic activity along the Min River, Eastern Tibet"

_Earth Surface Dynamics, 2022_

## Author Comment (AC1)

We are grateful to **Reviewer 1** for giving us detailed comments to improve our manuscript. The responses to the comments from Reviewer 1 are addressed point by point as follows.

Comments

Shi et al. present combined analyses of grain-size distribution and magnetic susceptibility in modern fluvial sediments from the upper Min River, Eastern Tibet to distinguish regional tectonic vs climatic influences on fluvial deposition. Their study indicates that tectonic faulting exerts a first-order control on fluvial deposition in tectonically active regions, which are well supported by regional relief and hillslope angle data. As far as I know, this study proposes a new and plausible method to characterize the development of tectonic activity along a transect, which is distinctly different from the traditional low-temperature thermochronologic dating and seismic methods as we often see. Therefore, I recommend this manuscript for publication. In the meantime, there are some minor issues that are needed to be clarified before this manuscript can be completely accepted.

Thank you very much. Revisions have been made below according to your comments.

**Comments**

P33-34: The occurrence of well-sorted and well-rounded sediments may be related to a significant increase in rainfall and runoff at the source area with more erodible bedrocks, or to long-distance transport which may experience multiple sediment recycling.

Yes, we've also considered the possibility you mentioned. The segment IV is located in the downstream of the Dujiangyan, and large amounts of coarse debris (>250 μm) was captured in Zipingpu Reservoir. Well-sorted and well-rounded (pebbles) of fluvial sediments in segment IV may be related to long-distance transport which may experience multiple sediment recycling, further to say that it must be related to the long-time of scouring and sorting by rivers. We have corrected it in line 33-34.

P109-110: Which analyses of river sediments should be specified here.

Yes, we have specified that in line 111.

P118-119: …and tectonic activity is tectonically controlled by….

Thanks, we have corrected it in line 120.

P141: The location of Fig. 1a was not shown in the inset map.

Thanks, we have marked it in the inset map.

P200: the bedrock is `naked` well exposed.

Thanks, we have corrected it in line 203.

P286: what does the dotted line represent at the upper left part (from the top to about No 50) of the SUS data?

The dotted lines represent the average value of the whole sequence. We have specified that in the figure 3 caption (line 289).

P435: a higher of rivers incision rates???

We mean that active tectonics activity forced a higher regional denudation rate, rather than the river incision rates. We have corrected it in line 439.

P488: seem a little contradict with???

We mean that the hillslope angles and local relief gradually increase downstream along the Minjiang Fault are decoupled with the high and stable proportion of fine–grained background dust in the fluvial sediments of segment I. We have corrected it in line 492-494.

At last, I personally think the clarity of Figs 3-7 could be improved significantly. It is better to mark in color mode.

Thanks, we have modified Figs 3-7.

---

## Author Comment (AC3)

We are grateful to **Reviewer 2** for giving us comments to improve our manuscript. The responses to the comments from Reviewer 2 are addressed point by point as follows.

Comments

The manuscript conducted a series of grain-size analysis for modern fluvial sediments from the upper Min River, Eastern Tibet and then tried to indicate the control of tectonic activity on fluvial sediments. Although the authors provided some data of grain-size distribution of the modern fluvial sediments along the upper Min River channel, the relationship between sedimentary grain size and tectonic activity is far away to be reach. In fact, as stated by the manuscript itself, there are many factors to determine the grain-size distribution of fluvial sediments, including climate, vegetation, hydrology, geomorphology, lithology, and fault slip rate etc. Thus, it is lack solid evidence for their implication.

First of all, thank you very much for your review.

Fluvial sediments are important materials for studying tectonic activity and climate change. Because fluvial sediments can reflect provenance and transport power. Grain-size distribution is an important parameter to characterize sediment. In this study, we selected fluvial sediments in the upper Min River, eastern Tibet, and discussed the main controlling mechanism of fluvial sediment provenance through detailed grain-size analyses. This paper innovatively uses grain-size of fluvial sediments to study the tectonic activity in eastern Tibet, and confirms the previous research results.

Although there are many factors to determine the grain-size distribution of fluvial sediments, including climate, vegetation, hydrology, geomorphology, lithology, and tectonic activity etc. But we believe that the dominant role of the grain-size distribution of fluvial sediments is unique. Based on the comprehensive consideration of various factors (vegetation, hydrology, geomorphology, lithology, and tectonic activity), this paper attempts to establish the ideal relationship between grain-size distribution and tectonic activity and/or climate along upper Min River in the eastern Tibetan Plateau, rather than discussing the relationship between fluvial sediments and tectonic activity in isolation.

Based on grain size parameters (grain–size fractions, EMs, and mode values, C-M, F-M, Y value) and existing knowledge of regional sediments (P346-405), we discussed the possible origin and indicative significance of EMs, and obtained the following insights: EM1 (fine–grained fractions, <63 μm) probably have an aeolian provenance (P347-366); EM2 mainly reflect typical fluvial sediments (P374-385); EM3 represents the local sedimentary component that was locally transported over short distances (P386-405).

According to the piecewise characteristic of grain-size components, combined with climate, vegetation, hydrology, geomorphology, lithology, and fault slip rate, we discussed the controlling mechanism for piecewise changes in grain size distribution along the upper Min River. Among them,

1) The windy and semi–arid climate in the study area is responsible for more fine particle components (EM1) in segment I, and Segment I developed along the Minjiang Fault (Fig. 1a), which has a low slip rate (0.30–0.53 mm/a) and therefore a weak influence on local provenance supply (P407-419).

2) Fluvial sediments coarsen at the transition between segments I and II, highlighting an increase in EM2 and EM3 content, and a higher M value (Figs. 3, 7). This locality occurs at intersection of the Minjiang Fault and the Songpinggou Fault (Fig. 1a), which was the epicenter of the 1933 Ms 7.5 Diexi earthquake. In addition, the longitudinal slope of the riverbed (12.6‰, Fig. 7c) and the hillslope angle (41.4°, Fig. 7d) are highest in the entire study area, which imply higher regional denudation rates forced by active tectonics (P426-445).

3) EM3 content rapidly reaches its maximum in segment III (Fig. 5), likely due to the maximum transport force (C value) in the area (Fig. 7). The continuous occurrence of the coarsest grain–size in the segment III responds to the high slip rate of the faults when the catchments cross the faults (P446-472).

4) Segment IV is located in the Sichuan Basin and is completely unaffected by alpine valleys in the eastern TP. In fact, the flurosion was the main factor controlling sediment transport (EM2) in segment IV (P420-424).

Based on multiple analyses, this paper establishes the ideal relationship between grain-size distribution and tectonic activity and/or climate along upper Min River in the eastern Tibetan Plateau,

Comments

1. No question was raised for solving in the Introduction, so I cannot catch the significance of the study. In fact, nearly all discussion or implications are common knowledge, without solid contribution.

It is a scientific issue expounded in the Introduction to obtain the regional tectonic activity through the comprehensive study of topography, vegetation, hydrology, deposition and regional geology.

The solid contribution of this study is to obtain regional tectonic activity through modern fluvial sediments combined with geomorphology, hydrology, vegetation and regional geology, which is distinctly different from the traditional low-temperature thermochronological dating, GPS and seismic methods as we often see.

2. There are coarse sediments and gravels along the Min River, but no data for these depositions.

Yes, the phenomenon you mentioned is ubiquitous along the Min River.

For example, the 2008 Mw 7.9 Wenchuan earthquake triggered the largest number of coseismic landslides on record (~200,000) and large quantities of debris (coarse sediments and gravels) over vast areas (>44,000 km$^2$) (Dai et al., 2011; Xu et al., 2014). From 2008 to 2020, these deposits can be remobilised by four rainstorms (13 Aug 2013, 10 Jul 2013, 20 Aug 2019 and 10 Aug 2020), while more than 70% of the debris is stabilised on the hillslopes (Dai et al., 2021) and will take more than 370 years to move out (Wang et al., 2017). The above results are sufficient to show that the extensive existence of gravel in Min River can reveal tectonic activity for a long time, and we can make use of it. But these studies cannot explain the regional fluvial and wind transport characteristics. Therefore, the modern river sediment samples were collected along the upper Min River (~340 km) to study the transport information of various forces, such

as hydrology, wind force and tectonics. And then, we report a new approach that can reveal the style of regional tectonic activity by analyzing fluvial sediments collected from tectonically active regions. That is the new innovation of this paper.

We have thoroughly revised grammar and type mistakes.

The manuscript was retouched twice before submission.

---

## Author Response (AR1)

**Response to Reviewer #1**

Shi et al. present combined analyses of grain-size distribution and magnetic susceptibility in modern fluvial sediments from the upper Min River, Eastern Tibet to distinguish regional tectonic vs climatic influences on fluvial deposition. Their study indicates that tectonic faulting exerts a first-order control on fluvial deposition in tectonically active regions, which are well supported by regional relief and hillslope angle data. As far as I know, this study proposes a new and plausible method to characterize the development of tectonic activity along a transect, which is distinctly different from the traditional low-temperature thermochronologic dating and seismic methods as we often see. Therefore, I recommend this manuscript for publication. In the meantime, there are some minor issues that are needed to be clarified before this manuscript can be completely accepted.

Thank you very much. Revisions have been made below according to your comments.

P33-34: The occurrence of well-sorted and well-rounded sediments may be related to a significant increase in rainfall and runoff at the source area with more erodible bedrocks, or to long-distance transport which may experience multiple sediment recycling.

Yes, we've also considered the possibility you mentioned. The segment IV is located in the downstream of Dujiangyan, and large amounts of coarse debris (>250 μm) was captured in Zipingpu Reservoir. Well-sorted and well-rounded (pebbles) of fluvial sediments in segment IV may be related to long-distance transport which may experience multiple sediment recycling, further to say that it must be related to the long-time of scouring and sorting by rivers. We have corrected it in lines 32-34.

P109-110: Which analyses of river sediments should be specified here.

Yes, we have specified that in line 110.

P118-119: …and tectonic activity is tectonically controlled by….

Thanks, we have corrected it in line 119.

P141: The location of Fig. 1a was not shown in the inset map.

Thanks, we have marked it in the inset map in Figure 1.

P200: the bedrock is naked well exposed.

Thanks, we have corrected it in line 202.

P286: what does the dotted line represent at the upper left part (from the top to about No 50) of the SUS data?

The dotted lines represent the average value of the whole sequence. We have specified that in the figure 3 caption (line 288).

P435: a higher of rivers incision rates???

We mean that active tectonics activity forced a higher regional denudation rate, rather than the river incision rate. We have corrected it in line 439.

P488: seem a little contradict with???

We mean that the hillslope angles and local relief gradually increase downstream along the Minjiang Fault are decoupled with the high and stable proportion of fine–grained background dust in the fluvial sediments of segment I. We have corrected it in lines 493-495.

At last, I personally think the clarity of Figs 3-7 could be improved significantly. It is better to mark in color mode.

Thanks, we have modified Figures 3-7.

**Response to Reviewer #2**

The manuscript conducted a series of grain-size analysis for modern fluvial sediments from the upper Min River, Eastern Tibet and then tried to indicate the control of tectonic activity on fluvial sediments. Although the authors provided some data of grain-size distribution of the modern fluvial sediments along the upper Min River channel, the relationship between sedimentary grain size and tectonic activity is far away to be reach.

I cannot agree with you.

Although this study focuses on grain-size analysis, we also analyzed the data including magnetic susceptibility, hillslope angle and relief, runoff and precipitation, bedrock, fault activity with frequent earthquakes, and vegetation.

The current grain-size analysis includes a variety of indicators, including the C-M and F-M diagrams to reveal the transport dynamics, end–member analysis to reveal the transport process, Y value to indicate the source characteristics, the grain size frequency distribution curve to distinguish aeolian and fluvial provenance characteristics, and the ultrafine component to correspond pedogenesis.

In addition, previous studies have shown that it is feasible to discuss the main control factors for changes in grain-size (Singh et al., 2007; Bravard et al., 2014).

In this study, based on a series of grain size parameters (grain–size fractions, EMs, and mode values, C-M, F-M, and Y value) and existing knowledge of regional sediments (lines 346-405), we discussed the possible origin and indicative significance of EMs, and obtained the following insights:

EM1 (fine–grained fractions, <63 μm) probably has an aeolian provenance (lines 347-366); EM2 mainly reflect typical fluvial sediments (lines 374-385); EM3 represents the local sedimentary component that was locally transported over short distances (lines 386-405). Based on the indicative significance of EMs, combined with the regional climatic and tectonic background, our data support that the relationship between sedimentary grain size (EM3) and tectonic activity is reliable.

In fact, as stated by the manuscript itself, there are many factors to determine the grain-size distribution of fluvial sediments, including climate, vegetation, hydrology, geomorphology, lithology, and fault slip rate etc. Thus, it is lack solid evidence for their implication.

Although many factors determine the grain-size distribution of fluvial sediments, but there are one or two mainly controlling factors in specific regions, which is worth exploring.

The eastern Tibetan Plateau is geomorphologically characterized by alpine valleys. The Min River valley is typically steep, narrow and deepening downstream with an incision depth of 300–1500 m (lines 148-151). The upper reaches of the Min River have a windy and semi-arid climate. The mean annual precipitation (MAP) ranges from 400 mm to 850 mm, and the highest instantaneous wind speed recorded in the study area is 21 m/s (lines 159-166). The Songpan-Diexi-

Wenchuan-Dujiangyan segments of the upper reaches of Min River show low vegetation coverage, with a small number of shrubs and herbs, and the bedrock is widely exposed (lines 186-211).

The fault slip rate in the eastern margin of the Tibetan Plateau is ~1 mm/a based on GPS data (Zhang et al., 2010). However, the historical and instrumental records show that the Minshan uplift zone is probably one of the most tectonically active units in China and even the world (Sun et al., 2018), because it has experienced 8 strong earthquakes (≥ 6.0) along its boundary faults since 1933. Frequent seismic activities caused widespread geomorphic landscape damage, and released a large source of fresh terrigenous debris. The sudden coarsening of grain-size in the Xinmocun and Lixian lacustrine sections is considered to be related to earthquakes (Jiang et al., 2014, 2017). Note that the coarsest grain-size (EM3), and the abnormally high values of magnetic susceptibility in Wenchuan-Dujiangyan segment (Fig. 3) can be explained by the change of provenance caused by the tectonic influence of granite bedrock.

According to the segment characteristic of grain-size components (Fig. 2), combined with climate, vegetation, hydrology, geomorphology, lithology, and fault slip rate, we discussed the mainly controlling mechanism for piecewise changes in grain size distribution along the upper Min River, and established the ideal relationship between grain-size distribution and tectonic activity and/or climate along upper Min River in the eastern Tibetan Plateau.

No question was raised for solving in the Introduction, so I cannot catch the significance of the study. In fact, nearly all discussion or implications are common knowledge, without solid contribution.

Our scientific question is how to identify the regional tectonic activity along the upper reaches of the Min River. But the alpine valleys in the eastern Tibetan Plateau reduce the preservation potential of Quaternary sediments and expose large areas of bedrock, which hinder the study of tectonic activity through the trenching. Fortunately, fluvial sediments are wide distributed. The solid contribution of this study is to obtain regional tectonic activity through modern fluvial sediments combined with geomorphology, hydrology, vegetation and regional geology, which is distinctly different from the traditional low-temperature thermochronological dating, GPS and seismic methods.

There are coarse sediments and gravels along the Min River, but no data for these depositions.

Yes, the phenomenon you mentioned is ubiquitous along the Min River. For example, the 2008 Mw 7.9 Wenchuan earthquake triggered the largest number of coseismic landslides on record (~200,000) and large quantities of debris (coarse sediments and gravels) over vast areas (>44,000 km$^2$) (Dai et al., 2011; Xu et al., 2014). From 2008 to 2020, four rainstorms occurred (13 Aug 2013, 10 Jul 2013, 20 Aug 2019 and 10 Aug 2020), but more than 70% of the debris is stabilized on the hillslopes (Dai et al., 2021) and will take more than 370 years to move out (Wang et al., 2017).

The above results are sufficient to show that the extensive existence of gravel in Min River can reveal tectonic activity for a long time, and we can make use of it. But these studies cannot explain the relatively fine fluvial and wind transport characteristics. Therefore, the modern river sediment samples were collected along the upper Min River (~340 km) to study the transport information of various forces, such as hydrology, wind force and tectonics. And then, we report a new approach that can reveal the style of regional tectonic activity by analyzing fluvial sediments collected from tectonically active regions. That is the innovation of this paper.

There is no rule to divide the river into four segments in the main text, but it was divided into three segments in Abstract and conclusion.

Sorry you make a mistake. In "2.2 Segmented Characteristics of the Min River", the rule to divide the river into four segments and the characteristics of each segment are described in detail (lines 178-211).

In abstract and conclusion, we introduce and summarize the four segments of the upper Min River. Based on the analysis of grain size parameters, hydrological, vegetation, topography and landforms, and regional tectonics data, we divide the upper reaches of Min River into four segments, but the tectonically active is only divided into three sections. Among them, we conclude that tectonic processes play a dominant role in the segments II and III. Therefore, segments II and III are classified as the most tectonically active section (lines 27-32, 548-551) in abstract and conclusion.

The three segments of tectonic activity along the upper Min River show that the Minjiang Fault, situated in the Minjiangyuan–Diexi segment (segment I), generally shows weak tectonic activity; two segments of the Maoxian-Wenchuan fault from Diexi to Wenchuan (segments II) and from Wenchuan to Dujiangyan (segment III) show enhanced phase of regional tectonic activity, although the segment from Dujiangyan to the Sichuan basin segment (IV) records almost no evidence of tectonic activity (lines 546-551).

For the sections of "Climate controlled finer grained fluvial sediments" and "Coarser grained deposits controlled by tectonism", there is no solid evidence for either.

The discussion in the sections "5.2 Climate controlled finer grained fluvial sediments" and "5.3 Coarser grained deposits controlled by tectonism" is based on the conclusions of section "5.1 Dynamic and provenance implications of fluvial sediments".

In section 5.1 (lines 349-409), we discussed the possible origin and indicative significance of EMs based on grain size indicators (grain–size fractions, EMs, mode values, C-M, F-M, and Y value), physical geographic conditions (wind, hydrology, vegetation, and topography), and existing knowledge of regional sediments, and obtained the following insights: EM1 (fine–grained fractions, <63 μm) probably has an aeolian provenance (lines 350-377), EM2 mainly reflects typical fluvial sediments (lines 378-389), and EM3 represents the local sedimentary component that was locally transported over short distances (lines 390-409).

Based on the indicative significance of EMs, combined with the regional climatic and tectonic background, we support that the understanding of sections "Climate controlled finer grained fluvial sediments" and "Coarser grained deposits controlled by tectonism" are reliable.

There is no new implication in the conclusion section, just for weak or enhanced tectonic activity, nothing with grain-size distribution of fluvial sediments.

Before our work, the tectonic activity of the ~340km upper Min River is unclear. Our studies reveal that the tectonic activity of the upper segment (from Minjiangyuan to Diexi) of the Min River is weak and the lower segment (from Diexi to Dujiangyan) is strong. Then, we propose that the segmentation of tectonic activities can be explored through grain-size distribution of fluvial sediments, combined with the climate, vegetation, hydrology, geomorphology, lithology, and fault slip rate etc. Therefore, the grain-size distribution of fluvial sediments, geomorphology (local relief

and hillslope) and geology (fault and lithology) data are auxiliary data and means, and our aim is to propose new methods for revealing tectonic activity in the region.

There are lots of grammar and type mistakes.

We have thoroughly revised grammar and type mistakes. The manuscript was polished twice before submission by a native speaker.

**Reference**

Dai, F.C., Xu, C., Yao, X., Xu, L., Tu, X.B., Gong, Q.M.: Spatial distribution of landslides triggered by the 2008 Ms 8.0 Wenchuan earthquake, China. J. Asian Earth Sci., 40, 883-895, https://doi.org/10.1016/j.jseaes.2010.04.010, 2011.

Dai, L.X., Scaringi, G., Fan, X.M., Yunus, A.P., Liu, Z.J., Xu, Q., Huang, R.Q.: Coseismic debris remains in the orogen despite a decade of enhanced landsliding. Geophys. Res. Lett., https://doi.org/10.1029/2021GL095850, 2021.

Sun, J.B., Yue, H., Shen, Z.K., Fang, L.H., Zhan, Y., Sun, X.Y. The 2017 Jiuzhaigou earthquake: A complicated event occurred in a young fault system. Geophys. Res. Lett. 45, 2230–2240. https://doi.org/10.1002/2017GL076421, 2018

Wang, W., Godard, V., Liu-Zeng, J., Scherler, D., Xu, C., Zhang, J.Y., Xie, K.J., Bellier, O., Ansberque, C., Sigoyer, J., Team, A.: Perturbation of fluvial sediment fluxes following the 2008 Wenchuan earthquake. Earth Surf. Process Land., 42(15), 2611-2622, https://doi.org/10.1002/esp.4210, 2017.

Xu, C., Xu, X.W., Yao, X., Dai, F.C.: Three (nearly) complete inventories of landslides triggered by the May 12, 2008 Wenchuan Mw 7.9 earthquake of China and their spatial distribution statistical analysis. Landslides, 11(3), 441-461, https://doi.org/10.1007/s10346-013-0404-6, 2014.

Zhang, P.Z., Wen, X.Z., Shen, Z.K., Chen, J.H.: Oblique, High-Angle, Listric-Reverse Faulting and Associated Development of Strain: the Wenchuan Earthquake of May 12, 2008, Sichuan, China. Annu. Rev. Earth Planet. Sci. 38, 353–382. doi:10.1146/annurev-earth-040809-152602, 2010.

---

## Author Response (AR2)

**Comments to the editor.**

   We are very grateful for the attention and patience to our manuscript. Yes, some readers may have similar questions about Reviewer #2. We have incorporated our response to Reviewer #2 key issues in the revised manuscript.

In fact, as stated by the manuscript itself, there are many factors to determine the grain-size distribution of fluvial sediments, including climate, vegetation, hydrology, geomorphology, lithology, and fault slip rate etc. Thus, it is lack solid evidence for their implication.

   In lines 47-51, we emphasize that "although many factors determine the grain-size distribution of fluvial sediments, but there are one or two mainly controlling factors in specific region; in addition, previous studies have shown that it is feasible to discuss the main control factors for changes in grain-size", which is the theoretical and practical basis for the smooth implementation of this research.

No question was raised for solving in the Introduction, so I cannot catch the significance of the study. In fact, nearly all discussion or implications are common knowledge, without solid contribution.

   In lines 112-116, we emphasize the significance of this study.

There are coarse sediments and gravels along the Min River, but no data for these depositions.

   In lines 130-133, we illustrate that earthquakes have a significant effect on regional detrital source supply, and lacustrine deposition has a significant response to regional earthquakes.

   In lines 484-487, on the basis of the understanding of 2008 Ms 8.0 Wenchuan earthquake, we determine that " the extensive existence of gravel in Min River can reveal tectonic activity for a long time, but these studies cannot explain the relatively fine fluvial and wind transport characteristics". These further illustrate the significance of the study of fluvial sediments in the upper Min River.

There is no rule to divide the river into four segments in the main text, but it was divided into three segments in Abstract and conclusion.

   In "2.2 Segmented Characteristics of the Min River", the rule to divide the river into four segments and the characteristics of each segment are described in detail (lines 185-216).

   In abstract (lines 22-34) and conclusion (lines 547-551), we mainly emphasize the segmented character of regional tectonic activity along the upper Min River.

For the sections of "Climate controlled finer grained fluvial sediments" and "Coarser grained deposits controlled by tectonism", there is no solid evidence for either.

   In lines 406-408, we emphasize that the discussion in the sections "5.2 Climate controlled finer grained fluvial sediments" and "5.3 Coarser grained deposits controlled by tectonism" is based on the conclusions of section "5.1 Dynamic and provenance implications of fluvial sediments"